# Cellular and Molecular Biology of Cancer Stem Cells of Hepatocellular Carcinoma

**DOI:** 10.3390/ijms24021417

**Published:** 2023-01-11

**Authors:** Kuo-Shyang Jeng, Chiung-Fang Chang, I-Shyang Sheen, Chi-Juei Jeng, Chih-Hsuan Wang

**Affiliations:** 1Department of Surgery, Far Eastern Memorial Hospital, New Taipei City 22060, Taiwan; 2Department of Hepato Gastroenterology, Linkou Medical Center, Chang-Gung University, Taoyuan City 33305, Taiwan; 3Postgraduate of Institute of Medicine, National Taiwan University, Taipei 10617, Taiwan

**Keywords:** cancer stem cells, hepatocellular carcinoma, surface markers, chemoresistance, radioresistance

## Abstract

Hepatocellular carcinoma (HCC) is one of the leading causes of cancer death globally. The cancer stem cells (CSCs) of HCC are responsible for tumor growth, invasion, metastasis, recurrence, chemoresistance, target therapy resistance and radioresistance. The reported main surface markers used to identify liver CSCs include epithelial cell adhesion/activating molecule (EpCAM), cluster differentiation 90 (CD90), CD44 and CD133. The main molecular signaling pathways include the Wnt/β-catenin, transforming growth factors-β (TGF-β), sonic hedgehog (SHH), PI3K/Akt/mTOR and Notch. Patients with EpCAM-positive alpha-fetoprotein (AFP)-positive HCC are usually young but have advanced tumor-node-metastasis (TNM) stages. CD90-positive HCCs are usually poorly differentiated with worse prognosis. Those with CD44-positive HCC cells develop early metastases. Those with CD133 expression have a higher recurrence rate and a shorter overall survival. The Wnt/β-catenin signaling pathway triggers angiogenesis, tumor infiltration and metastasis through the enhancement of angiogenic factors. All CD133+ liver CSCs, CD133+/EpCAM+ liver CSCs and CD44+ liver CSCs contribute to sorafenib resistance. SHH signaling could protect HCC cells against ionizing radiation in an autocrine manner. Reducing the CSC population of HCC is crucial for the improvement of the therapy of advanced HCC. However, targeting CSCs of HCC is still challenging.

## 1. Introduction

Hepatocellular carcinoma (HCC) is one of the leading causes of cancer death globally [1,2]. Despite significant advances in the studies of etiologies, molecular investigations, imaging studies and various treatments, the 5-year overall survival rate and disease-free survival rate remain disappointing [1,3]. The treatment options for those with advanced HCC are limited, and finding novel treatments is an urgent necessity.

The etiologies of HCC are multifactorial, including viral infections, such as hepatitis B virus (HBV) and hepatitis C virus (HCV); nonalcoholic steatohepatitis; excessive alcohol consumption or cigarette smoking; and environmental toxins [4,5,6]. Regardless of the etiologies, most HCCs are associated with liver inflammation, fibrosis and cirrhosis [5]. Liver cirrhosis is present in the majority of patients with HCC [5,6]. Furthermore, the genomic instability caused by liver inflammation or liver cirrhosis and some signaling pathways may affect both the initiation and progression of HCC [5,6,7].

Cancer stem cells (CSCs) are defined as a subpopulation of poorly differentiated cancer cells that can grow, regenerate and invade tissues [8,9,10]. They may appear due to mutation or genetic evolution, and they usually present characteristics similar to those of normal stem cells [9,10]. Studies have found that the CSCs of HCC are responsible for tumor growth, invasion, metastasis, recurrence, chemoresistance, target therapy resistance and radioresistance [8,9,10,11,12,13,14,15]. Other studies have found that, under the impact of chemotherapy or radiotherapy, non-CSCs can transform into CSCs, encouraging cancer recurrence and worsening clinical outcomes [16,17,18]. Although there is no universal CSC marker that can be used to identify the CSCs of HCC, studies have shown some of the main surface markers that can identify liver CSCs, including epithelial cell adhesion/activating molecule (EpCAM) (CD326), cluster of differentiation 90 (CD90), CD44 and CD133 [19,20,21,22,23,24,25,26,27,28,29,30,31,32,33,34,35,36,37,38,39]. The reported main molecular signaling pathways attending the pathogenesis of the CSCs of HCC include the Wnt/β-catenin, transforming growth factors-β (TGF-β), sonic hedgehog (SHH), PI3K/Akt/mTOR and Notch pathways [40,41,42,43,44,45,46,47,48,49,50,51,52,53,54,55,56,57,58,59,60,61,62,63,64,65,66,67,68,69,70,71,72,73,74,75,76,77,78,79].

In this paper, we review the current literature regarding these main surface markers and molecular signaling pathways and the crosstalk and interactions among them, along with the role of CSCs in mitigating the effects of current therapies.

## 2. Surface Markers of HCC Cancer Stem Cells

### 2.1. Epithelial Cell Adhesion/Activating Molecule

Epithelial cell adhesion/activating molecule (EpCAM), a cancer antigen, is one of the most frequently reported markers of CSCs of HCC [20,21,80]. Khosla et al. found that the expression of EpCAM increases during the development of HCC [21]. Compared with a normal liver, a significantly higher expression of “stemness genes” has been found in liver cirrhosis and EpCAM-positive HCC. This suggests that EpCAM-positive liver CSCs may appear both in liver cirrhosis and during the formation of HCC [21]. Some studies have identified liver CSCs with an overexpression of EpCAM in HCC in individuals with a history of HBV infection, suggesting that hepatitis B antigen (HBx) may upregulate EpCAM and CSC phenotypes [81,82]. EpCAM expression in HCC is also correlated with elevated levels of alpha-fetoprotein (AFP) [83]. Meta-analyses have shown that EpCAM is correlated with the poor differentiation of HCC [21,83]. There is also evidence that EpCAM can affect tumor recurrence [21,83]. Patients who are positive for EpCAM have a shorter survival rate than patients who are negative for EpCAM (85.7%, 51.3% and 46.2% vs. 86.2%, 86.2% and 82.3% at one, two and three years, respectively) [12]. Zeng suggested that a higher value of EpCAM staining of liver explants predicted HCC recurrence after liver transplantation [13]. In addition, the expression of EpCAM is higher in patients with HCC receiving preoperative chemotherapy, especially when using cisplatin treatment [82].

Yamashita et al. categorized HCC into subgroups according to EpCAM expression and AFP level [83]. Patients with EpCAM-positive and AFP-positive HCC are usually young but in advanced TNM stages. Those with EpCAM-negative and AFP-negative HCC are usually older but in earlier tumor-node-metastasis (TNM) stages [83]. EpCAM-positive and AFP-positive HCC may present the characteristics of hepatic stem/progenitor cells, whereas EpCAM-negative and AFP-negative HCC only has similar features to adult hepatocytes [83].

### 2.2. Cluster of Differentiation 90

Cluster of differentiation 90 (Th-1 cell surface antigen; CD90), a 25–37 kDa glycophosphatidylinositol-anchored cell surface protein, is another well-known HCC stem cell marker [22,23,24]. Yang et al. isolated CD90-positive HCC cell lines from both tumor tissues and blood samples, and they found that, compared with CD90-negative HCC cells, CD90-positive HCC cells have a higher malignant level with metastatic potential [24]. Further, Sukowati et al. reported that CD90-positive HCCs are usually poorly differentiated with a worse prognosis compared to the cells of cirrhotic liver or normal liver [22,84].

Guo et al. compared patients with CD90-positive HCC and those with CD90-negative HCC and found that the overall survival rates at one, two and three years were 87.5% vs. 100%, 72.9% vs. 94.1% and 54.7% vs. 88.2%, respectively [12]. Meta-analyses have also shown that the specificity of CD90 is about 91.9% and that the sensitivity is about 48.22% when predicting the poor differentiation of HCC [15].

### 2.3. CD44

CD44, belonging to the transmembrane glycoprotein family, can be found in monocytes and neutrophils [85,86]. It plays a pivotal role in various cellular processes [25,26,27,28,29]. CD44 and its variants can also be found in various cancers and tissues, including HCC, peri-HCC tissue, hepatoblastoma, liver tissue with viral hepatitis infection and even normal liver tissue [87]. CD44-positive HCC cells present the characteristics of CSCs [25,26,87]. In HCC, coexisting expressions of other CSC markers, such as CD133 and CD90, can usually be found in CD44-positive CSCs [28,37,87,88]. The role of CD44 in HCC still remains controversial. It affects the epithelial–mesenchymal transition (EMT) and enhances glutathione synthesis to interact with the glutaminecysteine transporter, thereby enhancing the antioxidant potential of CSCs [25].

Luo et al., using a meta-analysis, reported that CD44 expression correlates with more advanced stages of HCC with a worse 5-year survival [29]. The correlation between CD44 and cell differentiation grade, AFP level and disease-free survival has no statistical significance [29]. In addition, another meta-analysis showed a correlation between CD44 and poorly differentiated HCC [12,29,87]. In resected HCC specimen studies, the survival of individuals with HCC with a low expression of CD44 was better than that of individuals with HCC with a high expression of CD44 (mean 73.2 months vs. 44.84 months, respectively) [12]. In particular, one study showed that HCC expressing one isoform (CD44v6) had a high incidence of extrahepatic metastasis [89]. However, there is no correlation between CD44v6 expression and tumor characteristics, such as encapsulation, vascular invasion, tumor size and the grade of cellular differentiation [89]. Some investigators have found that a higher expression of CD44 is present in circulating HCC cells than in the primary HCC [19]. They also found that CD44-positive HCC cells develop metastases more rapidly in mice than CD44-negative HCC cells [19]. This evidence, therefore, suggests that the CD44 phenotype may play an important role in the metastasis of HCC [12,29].

### 2.4. CD133

CD133, a penta-span transmembrane protein, may play a pivotal role in membrane topology organization [36]. CD133 appears in the CSCs of various tumors and diseased livers, such as in HCV infection, but it is absent in normal liver tissue [30,31,33]. The activation of CD133 affects the growth of HCC [36,37].

Some studies have reported a significant correlation between CD133 expression and the clinicopathological characteristics of HCC, including tumor grade, tumor stage, AFP serum level, malignant potential, a higher recurrence rate and a shorter overall survival [31,32,33,34]. CD133, therefore, shows potential as an independent prognostic factor [33]. Song et al. compared patients with HCC with high CD133 and those with low CD133, finding that the 5-year overall survival rate was 19.23% vs. 50%, respectively [33].

## 3. Signaling Pathways Involved in Cancer Stem Cells of HCC

Recent studies have shown that the main molecular pathways involved in liver CSCs include the Wnt/β-catenin, TGF-β, SHH, P13K/Akt/mTOR and Notch

### 3.1. Wnt/β-Catenin

Abnormal β-catenin activity is correlated with HCC in individuals with HCV and HBV infections [42]. The activation of the Wnt/β-catenin pathway enhances the intracellular accumulation of β-catenin, and β-catenin could then translocate into the nucleus to trigger the transcription of *Wnt* target genes, such as *c-Myc* and matrix metalloproteinases (MMPs), in order to facilitate HCC progression [40,41,42,43,44,45]. Chen et al. reported a high expression of Wnt/β-catenin in HCC [40]. In upregulated Wnt/β-catenin CSCs, the overexpression of its target genes, proto-oncogene c-MYC and cyclin D1 also could be found in CD44-positive CSCs [90]. This overexpression facilitates proliferation and tumor sphere formation, and it increases the tumor growth [90]. The role of β-catenin and its gene mutation (*CTNNB1*) in HCC progression has been emphasized in some clinical studies. β-catenin signaling enhances the activity of cancer stem cells to promote HCC growth [47]. It may also modulate the expression of angiogenic growth factor [47,91]. Furthermore, EpCAM and β-catenin could develop a complex to activate proto-oncogene *c-MYC* and cyclins A and E [90].

Some cross-interactions occur among Wnt/β-catenin signaling and different components of the tumor microenvironment (TME) of HCC [91]. The TME of HCC includes immune cells, stem cells, tumor vasculature and noncellular components. The Wnt/β-catenin signaling pathway triggers angiogenesis, tumor infiltration and metastasis through the enhancement of angiogenic factors, such as matrix metalloproteinase-2 (MMP-2), matrix metalloproteinase-9 (MMP-9), basic fibroblast growth factor (bFGF), vascular endothelial growth factor (VEGF) and VEGF-C [91]. Furthermore, β-catenin protects HCC cells from apoptosis and enhances cell migration via the activation of the epithelial–mesenchymal transition (EMT) and the upregulation of MMP. Upstream mediators, including non-coding RNAs (ncRNAs), may regulate β-catenin signaling in HCC, which, in turn, may participate in mediating drug resistance and immunoresistance in HCC [91]. As a result, anti-cancer agents that inhibit β-catenin signaling and mediate its proteasomal degradation are a possible avenue for HCC therapy.

Mutations in the β-catenin gene 1 (*CTNNB1*) exist in about 20–40% of all patients with HCC [47]. Mutations of the *CTNNB1* gene encoding β-catenin and its overexpression could trigger the progression and migration of HCC [47]. Cieply et al. found that the presence of *CTNNB1* mutations is correlated with a larger tumor size and macrovascular or microvascular invasion [92]. Overall, CTNNB1 mutation causes aberrant β-catenin signaling to contribute to tumor aggressiveness [47,93].

A molecular classification of HCC into two groups based on Wnt-pathway aberrations in HCC was proposed by Lachenmayer: the CTNNB1 molecular class and the Wnt-TGF-β molecular class [93]. The Wnt-TGF-β class is correlated with the cytoplasmic accumulation of β-catenin, vascular invasion and a higher risk of early recurrence after surgical resection [93]. Cytoplasmic β-catenin expression is associated with poor histological differentiation, a tumor size over 5 cm and a shorter disease-free survival [93].

As a potential counter to this, investigators have found that CWP232228, a Wnt/β-catenin small-molecule inhibitor, inhibited CD133+/acetaldehyde dehydrogenase (ALDH)+ liver CSCs, possibly decreasing the self-renewal capacity of CSCs and suppressing tumorigenicity in vitro and in vivo [94].

### 3.2. Transforming Growth Factor β (TGF-β)

During the process from chronic inflammation to HCC, cancer cells grow in an environment enriched with extracellular matrix proteins. TGF-β may orchestrate the crosstalk between tumor cells and the host stroma, and they could also promote EMT [53,55,58,95].

In HCC cells, when TGF-β induces EMT, it may trigger a switch to express stem cell genes and their potential for stemness, migration and invasiveness [58,95]. However, the induction of EMT by TGF-β is probably only partial. In some epithelial HCC cells, EMT may enhance the mesenchymal genes and CD44 to maintain epithelial gene expression. In human HCC tissues, the expression of CD44 correlates with the overexpression of EpCAM and CD133, suggesting that the co-expression of both epithelial and mesenchymal stem-related genes could occur concomitantly [53,56,57,58,60]. Epithelial cells show a higher stemness potential than mesenchymal cells. The expression of TGF-β is associated with partial EMT augments, mesenchymal genes and CD44, and it maintains the activation of epithelial-related genes [95].

TGF-β could activate CD133 expressions and suppress the expressions of DNA methyltransferase (DNMT) 1 and DNMT 3 beta, which could maintain DNA in a methylated state [60]. The demethylation of CD133 promoter-1 in CD133-negative cell lines may cause the overexpression of CD133. TGF-β-induced CD133 cells have the potential to initiate tumor development in vivo [60].

### 3.3. Sonic Hedgehog (SHH) Signaling Pathway

The SHH signaling pathway (SHH) plays a pivotal role in the proliferation and invasiveness of HCC cells [48,49,50,51,52]. The activation of the SHH signaling pathway occurs after the HH ligand binds to its receptor Patched1 (PTCH1) [51,52]. Then, smoothened (SMO), the G-protein-coupled receptor (GPCR)-like signal transducer, spares the suppression by PTCH1. It is possible that the activated SMO inhibits the release of transcription-factor glioma-associated oncogene (GLI) 2/3. Subsequently, GLI binds to the promoters in the nucleus in order to initiate the transcription of target genes, such as CYCLINs (cell proliferation), BCL2 (antiapoptotic) and SNAIL (EMT induction) [51,52]. Jeng et al. reported that the activation of the SHH signaling pathway is found in CD133+ Hepa1-6 HCC mouse cells [49]. SHH also affects the tumor microenvironment of HCC [51]. Jeng et al. also reported that inhibitors of the SHH pathway inhibited the growth of HCC in mice [51,96]. The SHH signaling pathway has potential as a treatment target for HCC if using a combination of inhibitors and other therapies [52,97].

### 3.4. PI3-Kinase/AKT/Mammalian Target of the Rapamycin (P13K/AKT/mTOR) Signaling Pathway

The activation of the PI3K/AKT/mTOR signaling pathway plays a role in the invasiveness of HCC, and it is frequently detected in immunohistochemical analyses of patients’ HCC tissues [61,62,63,64,65,66,67]. Furthermore, the activation of mTOR is usually correlated with poorly differentiated or advanced-stage HCC, early recurrence after surgical resection and worse prognosis [66,67]. Furthermore, problems remain due to the rapid drug resistance against AKT and mTOR inhibitor treatment in HCC cells, both in vitro and in vivo, which may be attributed to the rapid outgrowth of CSCs [61,62,63,64,65,66,67]. Another possible mechanism of drug resistance is the activation of some closely related signaling pathways (such as the RAS/RAF/MEK/MAPK signaling pathway) [98]. In addition, the activation of P13K/AKT/mTOR signaling also plays a role in HCC irradiation resistance [99]. Targeting PI3K/AKT signaling is a potential strategy for cancer [65].

### 3.5. Notch Signaling Pathway

The Notch family consists of evolutionarily conserved genes that encode for single-pass transmembrane receptors participating in stem cell maintenance [68,69,70,71,72,73,74,75,76,77]. During embryonic development and adulthood, intracellular Notch signaling is essential for cell specification, the maintenance of progenitor cells and lineage commitment [45,70,71,72,75].

In mammals, the canonical Notch pathway includes four receptors (1, 2, 3 and 4) and two ligand families (jagged 1 and 2 and delta-like ligand (DLL) 1, 3 and 4) [74,75]. The interaction between ligands Notch (Jagged 1 and 2 and Delta-like (DLL-1, 3 and 4)) and receptors (Notch 1, 2, 3 and 4) may activate the Notch pathway, especially after the interaction between the Notch 1 receptor and its ligand JAG1 [74,75]. The Notch intercellular domain (NICD) could translocate into the nucleus to initiate the transcription of the Notch-targeted genes hairy and enhancer of split 1 (HES1) and the HES1-related (HESR1) families. These transcription factors trigger cell proliferation, differentiation and apoptosis [74,75,76,77].

Villanueva et al. found that Notch occurs in about one-third of cases of HCC [74]. Notch signaling plays pleomorphic roles in HCC, enhancing tumor growth, invasiveness and stem-cell-like properties [73,74,75]. The Notch signaling pathway is a key regulator of macrophage polarization in liver disease [70].

Among the Notch receptors, only Notch 3 is present in the liver tissue during the middle embryonic stage [79]. Compared with the human adult liver and with the mature Buffalo rat liver cell line, a higher expression of Notch 3 was found in the differentiation of stem/progenitor cells (fetal liver stem/progenitor cells) [79]. Compared with normal liver tissue, Notch 3 is the most highly upregulated Notch pathway gene in HCC tissues [68,72,74,75,77,78]. An abnormal accumulation of Notch 3 is found among 78% of early HCCs [68]. It could, therefore, be a specific therapeutic target for HCC. The Notch target gene HES5, as a driver gene to promote tumorigenesis with its interaction partner AKT, has both pro- and anti-tumorigenic functions in HCC [68]. It has demonstrated a negative feedback loop, downregulating pro-proliferative MYC-target lactate dehydrogenase A (LDHA) and ornithine decarboxylase 1 (ODC1) and suppressing the Notch target HES1 [68].

## 4. Crosstalk among Pathways

Crosstalk among signaling pathways can occur [43,45,47,56,100]. Either the Notch or the Wnt/β-catenin signaling pathways could increase the stemness characteristics of the CSCs of HCC [45]. After the intersection of the Notch and Wnt/β-catenin signaling pathways, the liver CSCs of HCC may be enhanced [45]. Following the activation of Notch and Wnt/β-catenin in CSCs, tumorigenicity and self-renewal could be facilitated by the overexpression of CD44, CD133 and CD90. The accumulation of β-catenin increases the expressions of Jagged 1 and β-catenin, suggesting a crosstalk between these two pathways. Notch1 is downstream of Wnt/β-catenin, and the activation of the Wnt/β-catenin pathway could also facilitate the Notch1 intracellular domain (NICD) [45]. Moreover, Notch1 negatively affects the modulation of Wnt/β-catenin signaling. Using lentivirus N1ShRNA to knockdown Notch1 could upregulate the active form of β-catenin [45]. Furthermore, Zhang reported that Notch3 could modulate the stemness of HCC cells after the inactivation of the Wnt/β-catenin pathway [71].

Steinway et al. reported that the network modeling of TGF-β signaling in the EMT of HCC involves the activation of the SHH and Wnt pathways [55].

## 5. The Co-Expression and Interactions of Liver Cancer Stem Cell Markers and the Tumor Microenvironment

The clinical significance of the co-expression and interaction among these different liver CSC markers has been studied. A significantly poor differentiation of HCC is found in individuals with liver CSC with the co-expression of CD44 and CD133 [87]. Comparing individuals with HCC with high values of co-expression of CD44 and CD133 and those with low values, the survival is significantly different [88]. Higher expressions of octamer-binding transcription factor 4 (Oct4) (a self-renewal gene) and ATP-binding cassette super-family G member 2 (ABCG2) (the well-known drug resistance gene) are found in patients with a high co-expression of CD90 and CD133 [101]. It is also possible that these two gene expressions influence each other. Some investigators have found that CD133 expression could be suppressed via the mTOR-AKT pathway following the downregulation of CD90 [64]. Moreover, the overexpression of CD90 could occur through the mTOR pathway after the overexpression of CD133 and CD24 [62,63,64,65]. Based on a mouse model, CD90-positive and CD44-positive cells are more prone to lung metastasis with a higher tumorigenicity, and the downregulation of the CD44 could, therefore, inhibit the growth and metastasis of tumors [100].

The tumor microenvironment may also affect the extent to which liver CSCs facilitate the progression and maintenance of HCC [52,91,102,103,104]. CD44 and CD90 exist not only in the liver CSCs but also in cancer-associated fibroblasts (one main component of tumor stromal cells), and tumor-associated macrophages (TAM) could trigger an increase in CD44-positive liver CSCs [103].

Jeng et al. reported that SHH plays a role in the tumor microenvironment of HCC, while CSCs affect the resistance to the treatment modalities of HCC [51].

## 6. Chemotherapy May Encourage Liver CSCs

Chemotherapy for HCC to improve survival is limited in its effectiveness [105,106,107,108]. The commonly used chemotherapy (usually monotherapy) for HCC consists of cisplatin, vincristine (VIN), 5-fluorouracil (5-FU) or doxorubicin. There is no significant difference in the survival rates among these drugs [105,106,107]. Treatment failure may be attributed to liver CSCs evading chemotherapy. Hu et al. found an increase in sphere formation and the cancer cell stemness of liver CSCs after carboplatin therapy [16]. Wada et al. found that, after cisplatin treatment, CD44 cells manipulate the glutamine–cysteine transporter to avoid apoptosis and regenerate HCC cells [108].

Both overexpressed EpCAM CSCs and activated Wnt/β-catenin signaling CSCs could contribute to 5-FU treatment resistance [109,110]. CD133-positive HCC cells also resist 5-FU and VIN therapies [111]. Compared with CD133-positive cells, the upregulated adenosine triphosphate-binding cassette (ABC) superfamily transporters (ABCB1, ABCC1 and ABCG) could amplify resistance to 5-FU and VIN [111]. Adenosine triphosphate-binding cassette (ABC) family G member-2 (ABCG2) affect the chemoresistance of CD90+ CD133+ CSCs [101]. Chen found that CD133/EpCAM-positive liver CSCs enhancing EMT and the SHH signaling pathway have the possibility to facilitate resistance to chemotherapy [112]. The increase in liver CSC markers after the impact of chemotherapy suggests that HCC cells may shift to CSCs [16].

An activated SHH signaling pathway also affects chemoresistance [113]. However, it is possible that Notch inhibition increases the sensitization of CD133-positive HCC cells to VIN and 5-FU [111]. One group of investigators pre-incubated the Huh7 cell line with a Notch inhibitor, g-secretase dual antiplatelet therapy (DAPT)(N-[N-(3,5-difluorophenoacetyl-1-alanyl]-S-phenylglycine t-butyl ester), prior to treatment with an IC_50_ dose of VIN or 5-FU [111]. Then, they isolated the CD133-positive cells to analyze the cell viability, apoptosis, migration, spheroid formation and expressions of genes and proteins, finding that Notch inhibition sensitized the HCC CD133-positive cells to VIN and 5-FU via enhancing B-cell lymphoma 2 (BCL2)-binding component (BBC3-3)-mediated apoptosis [111]. The Notch/HES1/BBC3 axis may also play a role in the resistance of CD133-positive cells to VIN and 5-FU [111]. Zhang et al. found that Saikosaponin-d could inhibit HCC cells and enhance chemosensitivity via the inhibition of GLI 1 SUMOylation (SUMO: small ubiquitin-like modifier) under hypoxia [114]. The inhibition of mTOR also could increase the chemosensitivity of HCC [115].

## 7. Sorafenib and Liver CSCs

Sorafenib, a multi-targeted receptor tyrosine kinase inhibitor used as a standard target therapy, may suppress various kinases (consisting of vascular endothelial growth factor receptor 2 (VEGFR2), RAS/RAF/MAPK/ERK signaling pathways, etc.). However, the single use of sorafenib or the combination use of sorafenib with some chemotherapeutic agents failed to significantly increase patients’ survival [116,117,118,119,120,121,122]. This failure may be attributable to the presence of liver CSCs with some other molecular mechanism with chemoresistance [14,123,124]. Liver CSCs are also an important subpopulation of liver CSCs contributing to the resistance to sorafenib therapy [124]. Kim et al. compared patients with a high-CD133-expression HCC and those with a low CD133 expression and found that the progression-free survival (PFS) after sorafenib therapy was 4 months vs. 5.5 months, respectively [37]. When comparing individuals with high co-expressions of CD133 and CD90 and those with a low expression of both markers, the PFS was 2.7 months vs. 5.5 months, respectively [37]. For individuals with a high co-expression of CD133 and EpCAM and a low co-expression of both markers, the PFS was 4.2 months vs. 7.0 months, respectively [37]. It is possible that the overexpression of the ATP-binding cassette transporter family member (ABCG2) in CD133-positive liver CSCs resists sorafenib [37,111,125]. Kim suggested that the genes CD133 and CD90 may predict the response to sorafenib therapy [37].

The activation of SHH signaling is another possible mechanism for CD133-related sorafenib resistance. The cells induce sorafenib resistance via their overexpression of ATP-binding cassette (ABC) C1 (ABCC1) transporter [37,115,125]. Chemotherapy could reduce the effectiveness of sorafenib therapy by enhancing the expression of the inflammatory cytokine gene (including IL1, IL8 and IL11) in liver CSCs to prolong the survival of CSCs [126]. Following glucose deprivation, CD133+ liver CSCs could enhance the uptake of glucose via the overexpression of glucose transporters (GLUT1 and GLUT3) through the IL-6/STAT3 pathway [127]. Suppressing this process may sensitize the cells and encourage sorafenib-induced apoptosis [127]. However, Fekir et al. found that, in a hypoxic state, sorafenib-resistant liver CSCs could alter glucose metabolism via the activation of pyruvate dehydrogenase kinase 4 (PDK4) to avoid oxidative phosphorylation [17]. Suppressing PDK4 and reactivating oxidative phosphorylation may increase the chemosensitivity of the HCC of the CSCs via mitochondria reactivation [17]. Thus, overall, an altered liver CSC metabolism may enhance resistance to sorafenib therapy [17].

Another possible mechanism for sorafenib resistance in CD133+/EpCAM+ CSCs is the Wnt/β-catenin signaling pathway [128]. Compared with CD133-/EpCAM cells, the overexpression of Src-homology 2 domain-containing phosphatase 2 (Shp2) occurs in sorafenib-resistant CD133+/EpCAM+ liver CSCs [128]. Shp2 could activate the nuclear translocation of catenin via the dephosphorylation of cell division cycle 73 (belonging to the Paf1/RNA polymerase II complex) [128]. The dephosphorylated complex may bind to catenin in order to avoid degradation by glycogen synthase kinase 3 and induce nuclear translocation in order to enhance liver CSC self-renewal [128]. The catenin signaling pathway could affect the expression of testis-associated highly conserved oncogenic long non-coding RNA (THOR) in EpCAM+ liver CSCs, leading to the inactivation of FH535, a catenin inhibitor [46]. Some studies have reported that the PI3K/Akt/mTOR pathway may enhance the growth of liver CSCs in sorafenib-resistant advanced HCC via AKT activation [61,62,98].

EpCAM+ cells, another important liver CSC population, also play a crucial role in affecting sorafenib therapy. Guan et al. found that, after sorafenib therapy, the activation of the tuberous sclerosis complex protein kinase β (TSC-AKT) cascade increased from 38.5% to 58.7% in EpCAM+ cells [129]. This avoids the inhibition of TSC2 via extracellular-signal-regulated protein kinase (ERK) to form the TSC1/2 complex. The complex enhances the mTOR pathway to activate AKT [129].

In those with CD133+ liver CSCs, NANOG-dependent genes (Yap1 and Igf2bp3) could inactivate TGF-β signaling via the cytoplasmic retention of phosphorylated SMAD3 (SMAD: mothers against decapentaplegic and *C. elegans* protein SMA) to suppress SMAD3 phosphorylation and activate the IGF2BP3/AKT/mTOR pathway [130].

mTOR pathway activation also contributes to sorafenib resistance in CD133+/EpCAM+ liver CSCs and CD44+ liver CSCs [61,62,98,131,132]. CD133+/CD44+ liver CSCs may also resist sorafenib [9,133]. However, when Gedaly et al. incubated CD133+/CD44 liver CSCs for 72 h with sorafenib, the suppression of the cell cycle was found in 39% of CD133+/CD44+ liver CSCs [133]. The AKT is activated by the autocrine stimulation of the TGF-β of CD44+ liver CSCs, and it is correlated with the mesenchymal characteristics of these cells (the overexpression of SNAI1 and Vimentin) [131,134,135]. TGF-β affecting sorafenib is limited to CD44+ liver CSCs; the association does not exist in EpCAM+ or CD90+ liver CSCs [131,132,134,135,136,137].

Trials of second-line multi-kinase therapy agents (such as regorafenib, cabozantinib and ramucirumab) are currently ongoing [138,139,140].

## 8. Radiotherapy May Enhance Liver CSCs

The irradiation of mesenchymal stem cells (IR-MSCs), which pre-exist in the tumor microenvironment, could promote CD133+ cells in HCC [141]. IR-MSCs facilitate the stemness maintenance of liver CSCs via the activation of the Wnt/β-catenin signaling pathway [141]. After co-culturing with IR-MSCs, the colony and tumor formation potential of liver CSCs may be enhanced [141]. IR-MSCs also could facilitate the Wnt expression of CSCs. The addition of the Wnt inhibitor in the culture medium may suppress the stemness maintenance of IR-MSCs [141]. Chen et al. found that SHH signaling could protect HCC cells against ionizing radiation in an autocrine manner [142]. There is also evidence that the knockdown of GLI-1 could reverse radioprotective effects [142]. Tsai et al., using a cyclopamine study, found that the combined use of a SHH inhibitor and radiotherapy may enhance the radiosensitivity of HCC cells and orthotopic HCC tumors [143].

Based on preclinical evidence, Bamodu indicated that phosphoinositide-dependent kinase-1 (PDK1) is an active driver of irradiation (IR) resistance following the activation of PI3K/AKT/mTOR signaling [99]. Cancer stemness signaling is activated, and DNA damage is suppressed. Projecting PDK1 targeting could be a putative enhancer of radiosensitivity and a potential new therapeutic approach for those with IR-resistant HCC [99].

Hong et al. suggested that a subpopulation of CD133+ liver CSCs could resist sublethal irradiation and enhance the invasion and migration of HCC cells [144]. They found that A distintegrin and metallopeptidase domain 17 (ADAM17) silencing in ADAM17/Notch signaling could significantly inhibit the invasiveness and migration of enriched CD133+ CSCs after irradiation [144]. Thus, combining radiotherapy with cellular- or molecular-targeted therapies may probably affect the sensitivity or resistance to irradiation therapy. Targeting CSCs or signaling pathway proteins may lead to novel combination modalities capable of overcoming radioresistance. The phenotypical characteristics and functions of the surface markers of CSCs and the main signaling pathways are summarized in Table 1 and Figure 1.

## 9. Challenges of Targeting Cancer Stem Cells of HCC and Future Perspectives

Reducing the CSC population of HCC is crucial for the improvement of therapy for advanced HCC [7,8]. Drug delivery systems (DDs) aimed at CSCs are increasingly under investigation [153]. However, targeting the CSCs of HCC presents three key challenges. The first challenge is that HCC itself has heterogeneity, which contributes to tumor progression [145]. Furthermore, the heterogeneity of the CSCs of the same HCC remains [84]. Examinations of the circulating biomarkers of those with HCC may be a further issue [19,154]. The second challenge is that the treatment of the tumor microenvironment is relatively complex, with pre-tumor fibroblasts in the tumor microenvironment continuing to recruit CSCs in order to promote intrahepatic metastasis [104]. The impact of the microenvironment upon CSCs needs more investigations. The third challenge is the coexistence of different tumor markers and the co-expression of different signaling pathways in the CSCs of individual HCCs. This requires further detailed examinations before personalized treatment.

## 10. Conclusions

The CSCs of HCC are responsible for growth, invasion, recurrence, drug resistance and radioresistance. Various surface markers and different signaling pathways affect the CSCs of HCC. There are co-expressions, interactions and crosstalk among them. Reducing the population of CSCs is crucial for the treatment of advanced HCC. However, challenging problems still remain.

## Figures and Tables

**Figure 1 ijms-24-01417-f001:**
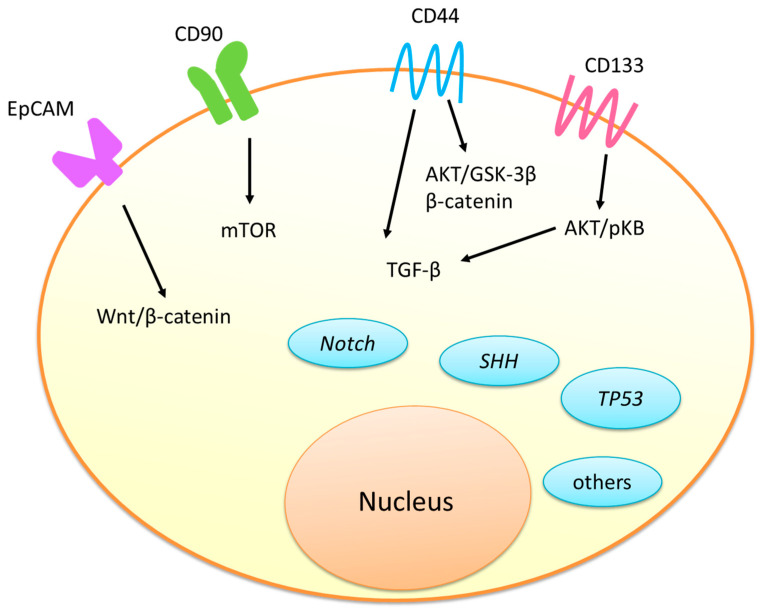
Signaling pathways involved in the cancer stem cells of hepatocellular carcinoma.

**Table 1 ijms-24-01417-t001:** Phenotypical characteristics and functions of the main cancer stem cell markers of hepatocellular carcinoma.

Phenotypical Characterists and Functions	EpCAM	CD90	CD44	CD133
Organogenesis	+			
Tumorigenesis	+		+	+
Self-renewal	+	+	+	+
Progression				+
Poorly differentiated	+	+	+	+
Early recurrence	+	+		+
Metastatic potential			+	
Shorter survival	+	+	+	+
Drug resistance	sorafenib	doxorubicin	doxorubicin	Doxorubicin 5-Fu sorafenib
Others	If AFP(+) young but advanced stage			
The possibly involved activating signaling pathway	Wnt/β-catenin	mTOR	Wnt/β-catenin TGFβ Akt/GSK-3β/β-catenin ERK/snail	Akt/pKB
References	[129,145,146]	[22,101,147]	[28,134,148,149]	[28,31,150,151,152]

## Data Availability

Not applicable.

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
