# Peer review of "Cellular and Molecular Biology of Cancer Stem Cells of Hepatocellular Carcinoma"

_ijms, 2023, doi:10.3390/ijms24021417_

Round 1

Reviewer 1 Report

The manuscript by Jeng et al. provides an overview of the main features of hepatocellular CSCs. Overall, the manuscript is well-organized and clear. However, some major points need to be addressed to improve the article:

1. Which are the phenotypical features of these CSCs? What about their metastatic potential or metabolic traits? A chapter about this topic should be added

2. Specific figures and tables summarizing the manuscript should be added

Author Response

Thank you very much for your comments.

1. Which are the phenotypical features of these CSCs? What about their metastatic potential or metabolic traits? A chapter about this topic should be added

We have added an paragraph “9. The phenotypical features of CSCs of HCC” We also have added a Table 1 to describe it.

2. Specific figures and tables summarizing the manuscript should be added

We have added Figure 1 and Table 1 to summarize the manuscript.

Reviewer 2 Report

Jeng et. al. reviewed cancer stem cell (CSC) markers in hepatocellular carcinoma (HCC). They described the main CSC markers in HCC and the main molecular signaling pathways. Also, they discussed chemotherapy, a tyrosine kinase inhibitor, and radiotherapy, as treatments for HCC. The authors concluded that reducing the population of CSCs is crucial but targeting CSCs is still a challenge. Few suggestions to improve the manuscript:

1-    The authors should add a table of the CSC markers with their characteristics and functions.

2-    The authors should add figures for the signaling pathways involved in CSCs to make it easier for readers to follow.

3-    The authors summarize the challenges of targeting CSCs for HCC. However, the authors should provide insights for future research directions to target CSCs.

Author Response

1. The authors should add a table of the CSC markers with their characteristics and functions.

We have added Table 1 to summarize the features and functions of these CSC markers.

2. The authors should add figures for the signaling pathways involved in CSCs to make it easier for readers to follow.

We have added Figure 1 to summarize the signaling pathways involved in CSCs of HCC.

3. The authors summarize the challenges of targeting CSCs for HCC. However, the authors should provide insights for future research directions to target CSCs.

We have added the parapgaph 10 “Challenges of targeting cancer stem cells of HCC and future perspectives” We have increased the future research directions in targeting CSCs in each challenges.

Round 2

Reviewer 1 Report

The manuscript can now be considered for publication in IJMS.

Author Response

(The authors gave the same response as above.)
